# Privacy-Preserving Object Detection with Secure Convolutional Neural Networks for Vehicular Edge Computing

**Tianyu Bai, Song Fu \* and Qing Yang \***

Department of Computer Science and Engineering, University of North Texas, Denton, TX 76203, USA
\* Correspondence: song.fu@unt.edu (S.F.); qing.yang@unt.edu (Q.Y.)

**Abstract:** With the wider adoption of edge computing services, intelligent edge devices, and high-speed V2X communication, compute-intensive tasks for autonomous vehicles, such as object detection using camera, LiDAR, and/or radar data, can be partially offloaded to road-side edge servers. However, data privacy becomes a major concern for vehicular edge computing, as sensitive sensor data from vehicles can be observed and used by edge servers. We aim to address the privacy problem by protecting both vehicles' sensor data and the detection results. In this paper, we present vehicle–edge cooperative deep-learning networks with privacy protection for object-detection tasks, named vePOD for short. In vePOD, we leverage the additive secret sharing theory to develop secure functions for every layer in an object-detection convolutional neural network (CNN). A vehicle's sensor data is split and encrypted into multiple secret shares, each of which is processed on an edge server by going through the secure layers of a detection network. The detection results can only be obtained by combining the partial results from the participating edge servers. We have developed proof-of-concept detection networks with secure layers: vePOD Faster R-CNN (two-stage detection) and vePOD YOLO (single-stage detection). Experimental results on public datasets show that vePOD does not degrade the accuracy of object detection and, most importantly, it protects data privacy for vehicles. The execution of a vePOD object-detection network with secure layers is orders of magnitude faster than the existing approaches for data privacy. To the best of our knowledge, this is the first work that targets privacy protection in object-detection tasks with vehicle–edge cooperative computing.

**Keywords:** data privacy; autonomous vehicles; deep learning; edge computing





## 1. Introduction

Autonomous vehicles (AVs) have been attracting more and more attention and interest in both industry and academia. AVs rely on various sensors, e.g., camera, LiDAR, radar, GPS, IMU, etc., to perceive the surrounding environment and plan movement and routes [1]. To achieve autonomous driving, objects on the road should be detected accurately and quickly. The latest object-detection methods or systems mostly use deep learning for detection. Although they are more accurate, deep-learning networks are compute-intensive and require powerful computing capacity on a vehicle. Furthermore, object-detection networks are only one of the many deep-learning networks that are run on a vehicle for various autonomous driving tasks.

To provide reliable computing power for delay-sensitive applications, edge computing [2] offers a cost-effective and scalable way to execute part of those deep-learning workloads for nearby vehicles. This *vehicle–edge cooperative computing* is attractive and practical for both existing ego AVs and future connected AVs.

Privacy, however, is a big concern in vehicle–edge collaboration, as the sensor data containing sensitive information leave the vehicle and are processed on an edge server. Both the sensitive information in the input data and the object results from the detection network can be accessed and used by the edge server. Existing approaches, such as homomorphic

encryption [3,4], secure multi-party computation [5], and secret sharing approximation [6], protect data privacy at the price of a prohibitive computational overhead and/or comprised detection accuracy.

To address these issues, in this paper, we present vehicle–edge cooperative deep learning with privacy preservation for object-detection tasks, named vePOD for short. vePOD aims to protect the privacy of both the sensor data from vehicles and the object results generated by CNN from being exposed to and used by edge servers which execute the CNN inference jobs. In vePOD, we leverage the additive secret sharing theory to design secure functions for every layer in an object-detection CNN. A vehicle's sensor data (e.g., a 2D image from an onboard camera) are partitioned and encrypted into two or more secret shares. Each share is transferred to and then processed on an edge server, which runs the object-detection CNN with secure functions on the secret share. The generated result from each edge server is then sent to the original vehicle (for workload offloading) or another vehicle (for workload offloading and data sharing), where the data are combined to obtain the detected objects. The entire process is configurable. When the vehicle–edge communication latency is high, a vehicle can help by executing the first several layers in the detection network and then send the secret shares of feature maps to edge servers for the remaining compute-intensive processing. This vehicle–edge cooperative computing paradigm can protect data privacy, reduce communication cost, and improve application performance.

We present two use cases where the layers in object-detection CNNs—Faster R-CNN (two-stage detection) and YOLO (single-stage detection)—are enhanced by secure functions in vePOD. The vePOD version of the networks, denoted by vePOD Faster R-CNN and vePOD YOLO, are tested on the COCO dataset. Experimental results show that vePOD Faster R-CNN and vePOD YOLO achieve the same detection accuracy as Faster R-CNN and YOLO, respectively. Secure functions on secret shares protect data privacy, but prolong the detection time. The execution time of vePOD Faster R-CNN is extended from 14 s (Faster R-CNN) to 44.1 s (vePOD Faster R-CNN) without any optimization and vePOD YOLO is extended from 3.2 s to 80.7 s. Compared with a slowdown of four orders of magnitude with homomorphic encryption on CNN [7], vePOD achieves a significant speed-up, which makes vePOD practical for real-world applications.

The rest of the paper is organized as follows. Section 2 describes object-detection CNNs and the detailed design of vePOD. Section 3 evaluates the performance of vePOD CNNs. The related research is presented and discussed in Section 4. Section 5 concludes the paper with remarks on future research.

## 2. Materials and Methods

### 2.1. Convolutional Neural Networks for Object Detection

AVs continuously collect real-time sensor data from the surrounding environment. These sensor data are processed to achieve various autonomous driving tasks. Among them, object detection is fundamental. Objects on the road, such as vehicles, pedestrians, cyclists, and other obstacles, must be detected accurately and quickly so that planning and control can drive a vehicle safely. While some AVs have expensive LiDAR sensors, cameras are the de facto sensors equipped on almost all AVs. Without loss of generality, we focus on object detection on 2D images in this paper. VePOD can be easily extended to work on 3D point clouds.

Object detection is a popular research topic. Many computer vision-based and deep learning-based algorithms have been proposed in the literature, such as [8–12]. Th recent advance of convolutional neural networks (CNN) promotes object detection. Two-stage and single-stage object-detection networks have emerged. *Two-stage object detectors* perform the detection task in two sequential steps: (1) locating the potential objects using bounding boxes, which are called region proposals; and (2) processing the region proposals with a prediction network to calculate classification scores, as well as locations. Examples of two-stage object-detection algorithms include R-CNN [13], Fast R-CNN [14], and

Faster R-CNN [15]). In contrast, *single-stage object detectors* scan an image only once to produce both the classification and bounding boxes of objects. YOLO [16] and SSD [11] are such algorithms.

In this paper, we aim to protect sensitive data from vehicles, sensor data and detected objects in particular, from being exposed to edge servers where object-detection CNNs are executed on the vehicle data. We design vePOD to be generic and applicable to both two-stage and single-stage object-detection networks. The case studies in Sections 2.5 and 2.6 explain how vePOD is used to enhance object-detection CNNs for privacy preservation.

### 2.2. Additive Secret Sharing Theory

The additive secret sharing theory provides a theoretical foundation for us to design the secure functions to process encrypted data shares from a vehicle in a vePOD object-detection network.

The concept of secret sharing was originally proposed to protect key security [17]. Informally, secret sharing works as follows. A trusted party divides a secret into $N$ shares, which are kept by $N$ participants. No less than $k$ ($k \leq N$) participants can reconstruct the secret. The major operations on secret shares are *divide* and *assemble*, as expressed in Equation (1).

$$
\begin{aligned}
D(\overrightarrow{s}) &= (\overrightarrow{s}_1, \overrightarrow{s}_2 \ldots \overrightarrow{s}_n), \ \overrightarrow{s} \in V \\
A(\overrightarrow{s}_1, \overrightarrow{s}_2 \ldots \overrightarrow{s}_n) &= \overrightarrow{s}
\end{aligned}
\tag{1}
$$

The additive secret sharing theory [18] employs additional constrains, that is, (1) secret shares maintain an additive relation; i.e., $s = \sum_{n=1}^{k} s_n$; and (2) $k = N$; i.e., a secret $s$ can be rebuilt only when all the secret shares are collected. Additive secure sharing enables secure operations on cipher text. For example, secrets $t$ and $s$ are divided into $(t_1, t_2, \ldots, t_n)$ and $(s_1, s_2, \ldots, s_n)$ among participants; then, the sum of secrets can be calculated without revealing their values; i.e., addition can be performed on secret shares separately, which can then be added up.

Based on the additive secret sharing theory, we develop three secure protocols for linear operations that are used in vePOD, i.e., secure addition (SecAdd), secure subtraction (SecSub), and secure scalar multiplication (SecSMul). An input array $(u, v)$ to SecAdd is split into $(u_1, v_1)$ and $(u_2, v_2)$, and sent to edge servers $S_1$ and $S_2$, respectively. Independently, $S_1$ calculates $f_1 = u_1 + v_1$ and $S_2$ calculates $f_2 = u_2 + v_2$. The addition of $f_1$ and $f_2$ produces the same result as the original addition; i.e., $f_1 + f_2 = u + v$. Similarly, an input array $(p, q)$ to SecSMul is divided into $(p_1, q)$ and $(p_2, q)$; edge servers $S_1$ and $S_2$ compute $f_1 = q * p_1$ and $f_2 = q * p_2$, respectively. We have $f_1 + f_2 = p * q$.

### 2.3. Overview of vePOD

vePOD aims to (1) protect a vehicle's sensor data from being exposed to or used by edge servers, and (2) protect the object-detection results from being obtained by edge servers. To achieve the first goal, a vehicle creates $N$ randomized secret shares $\{R_i\}$ from an image (or a feature map) $M$ following an additive relation, that is, $M = \sum R_i$. Each edge server processes one secret share, which contains random data. To protect against possible attacks in the public network between a vehicle and edge servers, a secret share $R_i$ is encrypted using an encryption key $K$ retrieved from a trusted server $T$, before being sent to an edge server. For the second goal, an edge server runs the vePOD CNN on a secret share $R_i$ and produces only partial results. We assume there is no conspiracy attack, that is, the $N$ edge servers do not exchange secret shares or partial results.

Each edge server executes an object-detection CNN on the received secret share. Figure 1 plots the workflow. The key challenge is *how to design an object-detection CNN that can process secret shares containing random data and still generate correct (partial) detection results*. To tackle this challenge, we examine every layer in an object-detection CNN (e.g., convolutional, activation, pooling, fully-connected layers, and others), and design secure

functions for each layer to handle secret shares. The design details of vePOD will be described in the next section.

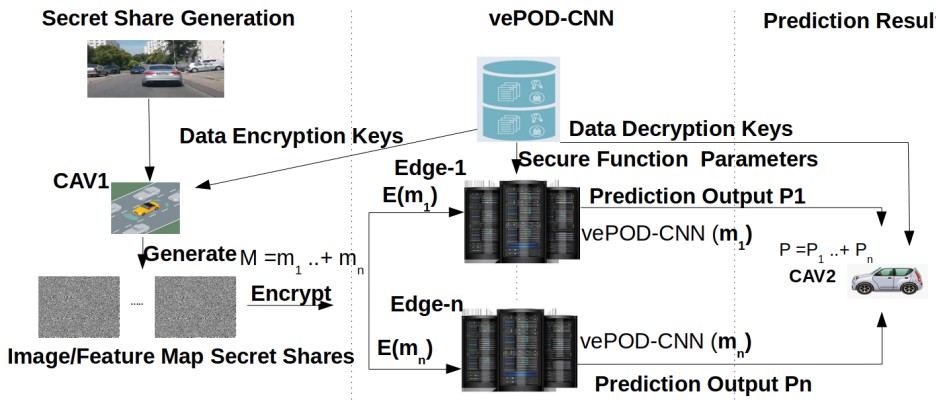

**Figure 1.** Workflow of vePOD: vehicle–edge collaborative deep learning with privacy preservation for object detection.

To illustrate where and how vePOD can be applied, we present an application scenario. Please note that this is one of many scenarios where vePOD can be used to protect a vehicle's data privacy. In this scenario, two or more vehicles exchange sensor data to expand their perception range and enhance perception accuracy. In cooperative perception, a vehicle needs to process not only its only sensor data, but data from other vehicles as well. To avoid overloading a vehicle, the sensor data from another vehicle can be offloaded to nearby edge servers, in which vePOD can be applied to protect data privacy while processing the vehicle's data on the edge. To apply vePOD, a vehicle partitions its sensor data to create secret shares, which are offloaded to edge servers. Each edge server runs a secret share through the secure layers in a vePOD CNN. The outputs from participating edge servers are combined on the destination vehicle to obtain the object-detection results. In this way, the privacy of the vehicle's sensor data is protected in both network transportation and edge computation.

### 2.4. Design of vePOD Deep-Learning Networks

In a vehicle–edge cooperative computing environment, both vehicles and edge servers participate in executing the object-detection CNNs, aiming to accurately detect objects around a vehicle and, at the same time, protect the vehicle's data privacy.

Recent studies [19–23] show that cooperative inference can protect input data by dividing a deep-learning network into two parts: one that resides on an end device and another that is on an edge server. The end device and edge server each execute part of the network and exchanges feature maps. Such a cooperative computing paradigm can better utilize the resources on both sides and improve the inference performance.

vePOD leverages this vehicle–edge cooperative computing paradigm to protect data privacy for vehicles. Our goal is to achieve privacy preservation with a high efficiency and uncompromised accuracy.

To facilitate our discussion, we use Faster R-CNN and YOLO, two widely used networks, as illustrative examples to describe the design of the vePOD CNN.

Faster R-CNN [15] is a two-stage object-detection network. It scans an input image twice. The first scan determines region proposals and the second classifies objects in those regions. The major components in the first stage include a backbone CNN to extract features, a region proposal network (RPN) to generate bounding boxes, and non-maximum suppression (NMS) to select bounding boxes from many overlapping ones. The second stage consists of region of interest (ROI) pooling to produce fixed-size feature maps for each bounding box and a prediction network to output detection results based on the feature maps and region proposals.

YOLO [16] is a single-stage detection network that treats object-detection as a regression problem and scans input images only once. The YOLO darknet network consists of 24 convolutional layers, 4 max-pooling layers, and 2 fully connected layers.

Image-Level and Feature-Level Secret Sharing

An important design decision that we need to make is on what data to generate secret shares. This can be done either on images or feature maps.

For image-level secret sharing, a vehicle generates multiple secret shares from an image. Edge servers execute object-detection CNNs on those secret shares, including feature map generation using a secure backbone CNN, region proposal prediction on encrypted data, secure ROI pooling, and a secure detection network. The advantage of this design is that the entire object-detection task is offloaded to edge servers, which reduces the vehicle's load. However, processing secret shares on multiple edge servers may cause RPN to produce erroneous region proposals, as RPN needs the original image to identify bounding boxes and refine the coordinates. Image-level secret sharing is effective for single-stage detection networks, as all the layers in a network are capable of processing individual secret shares without region proposals and global synchronization.

In contrast, feature-level secret sharing leverages region proposal generation on the vehicle. A vehicle runs RPN and creates secret shares based on the feature maps generated by the backbone CNN. The secret shares are then sent to and processed by edge servers; i.e., they go through a secure ROI pooling and a secure detection network. In feature-level secret sharing, the vehicle performs part of the object-detection network, including the backbone CNN and RPN, which leads to extra workload to the vehicle. However, the detection time can be reduced since the secure backbone CNN is slower than the original backbone network. Additionally, since only the secret shares of feature maps are exchanged between vehicle and edge servers, data privacy of the vehicle's images is protected, even under conspiracy attacks [24].

Which one, image-level or feature-level secret sharing, should be chosen for a given object-detection CNN? To answer this question, we need to consider the following factors: (1) the available resources on a vehicle and an edge server—when the vehicle has extra computing resources available, feature-level secret sharing is a better choice, as local execution avoids the additional overhead of running expensive secure functions; (2) the communication latency between vehicle and edge server—when the communication cost of transferring secret shares is lower than on-vehicle computation, image-level secret sharing can reduce the vehicle's load and better utilize edge resources; (3) The functional correctness of the object-detection network under different secret sharing schemes.

Next, we use two widely used object-detection CNNs, i.e., Faster R-CNN (a two-stage detection network) and YOLO (a single-stage detection network), to explain the construction of vePOD networks and the design of secure functions.

*2.5. vePOD Faster R-CNN: Vehicle–Edge Cooperative Object Detection for Privacy Preservation*

A region proposal network (RPN) iterates through a feature map and generates nine anchor boxes in three different aspect ratios and scales for each anchor point. Then, the bounding box regressor layer and classifier layer determine the shifted location of each anchor box and whether it is in the foreground or background with a probability score. The non-maximum suppression (NMS) groups anchor boxes based on an intersection over union (IOU) threshold. NMS selects the one with the highest probability score from each group. The 2000 filtered anchor boxes, together with their probability scores, are saved as region proposals. Those selected region proposals are projected to the feature map based on their coordinates. The projected regions on the feature maps are defined as the regions of interest (ROIs). ROIs are detected by the RPN based on the values in the feature map. The generated ROIs go through ROI pooling and a prediction network to detect objects.

The RPN and NMS need the whole image to find bounding boxes and refine their coordinates. As a result, image-level secret sharing causes the RPN to generate incorrect

bounding boxes, as each edge server only possesses a secret share of the original image. Therefore, we adopt feature-level secret sharing in the design of vePOD Faster R-CNN.

Figure 2 depicts the overall structure of vePOD Faster R-CNN and its key components. Both the vehicle and edge servers are involved in executing the object-detection network. On a vehicle, an input image goes through a backbone CNN and RPN; then, the generated feature maps are partitioned and encrypted. After receiving a feature-level secret share from the vehicle, an edge server runs an ROI pooling and a detection network on the secret share along with the region proposals. In order to produce correct detection results while processing encrypted secret shares, vePOD enhances the ROI pooling and detection network by introducing secure functions.

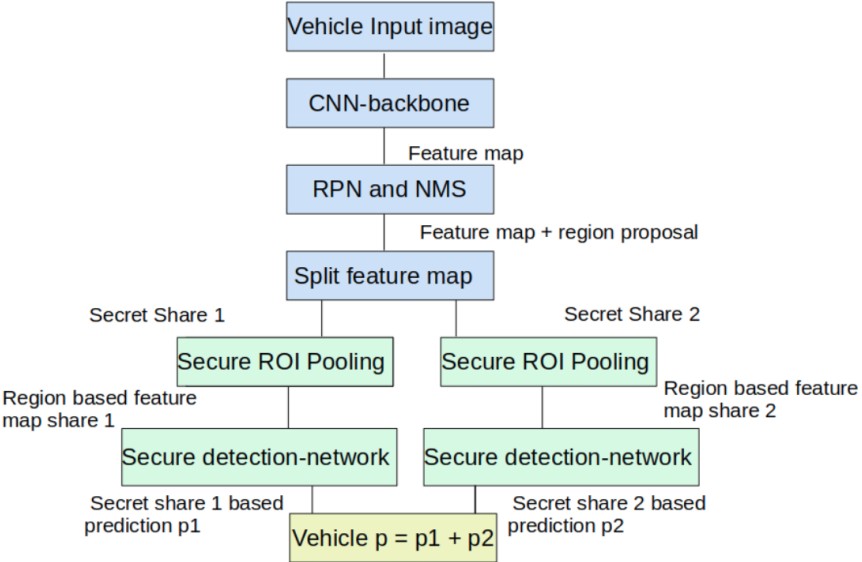

**Figure 2.** Structure and major components of vePOD Faster R-CNN. Blue blocks represent the operations performed on a vehicle and green blocks are those executed on an edge server; the yellow block can be done on the same or a different vehicle.

The detection network consists of two fully-connected (FC) layers. FC performs linear transformations on the input feature maps. Since a linear transformation does not change the additive relationship (Section 2.2) between secret shares, it does not affect the functional correctness when combining results from all secret shares. As a result, the secure function for the detection network is the same as the original detection network in Faster R-CNN.

In contrast, the ROI pooling performs non-linear transformations on input feature maps, which can change the additive relationship among the secret shares. The design of its secure function needs to address this problem.

### 2.5.1. Secure ROI Pooling

A region of interest (ROI) pooling generates fixed-size feature maps for each bounding box so that the detection network can classify the object in each bounding box.

The following operations are performed by the ROI pooling: (1) extracting region proposal specific feature maps; (2) dividing the extracted feature maps into a fixed number of patches based on the shape of the expected output feature map; and (3) selecting the maximum-valued element from each patch.

The secure ROI pooling takes feature-level secret shares and region proposals as its input from a vehicle. The operation (3) in the ROI pooling, i.e., selecting the maximum-valued element from each patch, involves non-linear computation (i.e., maximum). We develop a secure function to make it work correctly on a secret share.

Our approach is to compare the relative difference between secrets. For example, for two secrets V and D. which are partitioned into secret shares $(V_1, V_2)$ and $(D_1, D_2)$,

where $V = V_1 + V_2$ and $D = D_1 + D_2$, the secure function aims to determine whether $V$ is greater than $D$ without a reconstruction of the secrets. To achieve this, it calculates the difference $\eta = V - D = (V_1 + V_2) - (D_1 + D_2) = (V_1 - D_1) + (V_2 - D_2)$, and checks if $\eta$ is greater than zero. For each extracted feature-map patch, the secure function calculates $\eta$ for each pair of elements. By collecting those $\eta$'s from edge servers, we can know the relative difference of each secret, from which the maximum-valued element in secret shares can be identified.

Algorithm 1 sketches the secure function for ROI pooling. In the secure ROI pooling, each patch is flattened to a one-dimensional array with size $L_{ij} = width(P_{ij}) * height(P_{ij})$. Then, it creates a table $T_{ij}$ with size $(L_{ij})^2$ and computes $T_{ijmn} = P_{ij}[m] - P_{ij}[n]$. Table $\{T_{ij}\}$ stores the relative difference $\eta$ of $P_{ij}[m]$ and $P_{ij}[n]$ on an edge server that has the $i_{th}$ secret share and the $j_{th}$ patch. Edge servers $E_i$ and $E_q$ exchange $T_{ij}, T_{qj}$ through a trusted server and calculate $T_j = T_{ij} + T_{qj}$. $T_j[m][n]$ is the relative difference of the feature map $F$ on patch $j$, i.e., $T_j[m][n] = P_j[m] - P_j[n]$. If all the values in row $k$ are positive, then the corresponding value in $P_j$ is greater than the others.

As the number of secret shares $n$ increases, the secure ROI pooling scales well as $T_j = \sum_{n=1}^{N} T_{nj}$.

---

**Algorithm 1** Secure Function for ROI Pooling

---

1: **for** each region proposal **do**
2:     Scale the proposal by a scaling factor: $SP_i = P_i * \gamma$
3: **end for**
4: Extract region-based features $F$ with the scaled proposal's coordinates $\mu, v$
5: Calculate a width stride = $width(\kappa)/w$ and a height stride = $height(\kappa)/h$
6: Divide $\kappa$ into patches $P_{ij}$ based on the width stride and height stride
7: **for** each patch $P_{ij}$ **do**
8:     Flatten $P_{ij}$, the size of path $L_{ij} = width(P_{ij}) * height(P_{ij})$
9:     Create relative difference table T with a size $L_{ij} * L_{ij}$
10:     $T_{ijmn} = P_{ij}[m] - P_{ij}[n]$
11:     $E_i$ shares relative difference table $T_{ij}$ with $E_q$, collect relative difference table $T_{qj}$, calculate $T_j = T_{ij} + T_{qj}$
12:     **for** each row in $T$ **do**
13:         **if** all values in the row are greater than 0 **then**
14:             return the index of the row, $k$
15:         **end if**
16:     **end for**
17:     return $P_{ij}[k]$
18: **end for**

---

2.5.2. Selective Secure ROI Pooling

The secure ROI pooling (Section 2.5.1) leverages the additive secret sharing theory and a relative difference approach on feature maps to perform ROI pooling on encrypted secret shares by multiple edge servers. We note that although secret shares generated from a feature map contain random values, a comparison of the index of a maximum-valued element instead of calculating the difference can improve the efficiency of the secure function.

Specifically, after a single path is flattened into a single dimension, modern programming languages provide an application programming interface to retrieve the maximum-valued element's index. By comparing this index with those from other edge servers, the secure function can tell whether other secret shares have the maximum-valued element or not. This avoids the computation and construction of the relative difference table.

For example, the probability of two shares on patch $P_j$ having their maximum-valued element in the exact location is $\frac{1}{(S_{ij})^2}$. With two secret shares, this probability is relatively

high, justifying the feasibility of this optimization. As the number of secret shares increases, this probability drops and the effectiveness of the optimization is reduced .

### 2.6. vePOD YOLO: Privacy Preserving Object Detection on the Edge

YOLO is a single-stage object-detection network composed of multiple convolutional, max-pooling, and fully connected layers. The original image is not needed to generate region proposals. As a results, the image-level secret sharing can be used to build vePOD YOLO.

Figure 3 depicts the structure of vePOD YOLO and its major secure functions. In vePOD YOLO, a vehicle partitions an image into several shares and encrypts them to generate secret shares using an encryption key obtained from the trusted server. Those secret shares are distributed to edge servers, each of which executes vePOD YOLO consisting of secure functions of the convolutional, max pooling, and fully connected layers (detailed in the following sections) on a secret share. The outputs from vePOD YOLO on the edge servers are combined on a vehicle to obtain the detected objects. Compared with the feature-level secret sharing, the image-level secret sharing approach simplifies the processing on a vehicle and makes an edge server execute the entire secure detection CNN. This may lead to a longer execution time from running the secure functions that contain addition operations on secure shares.

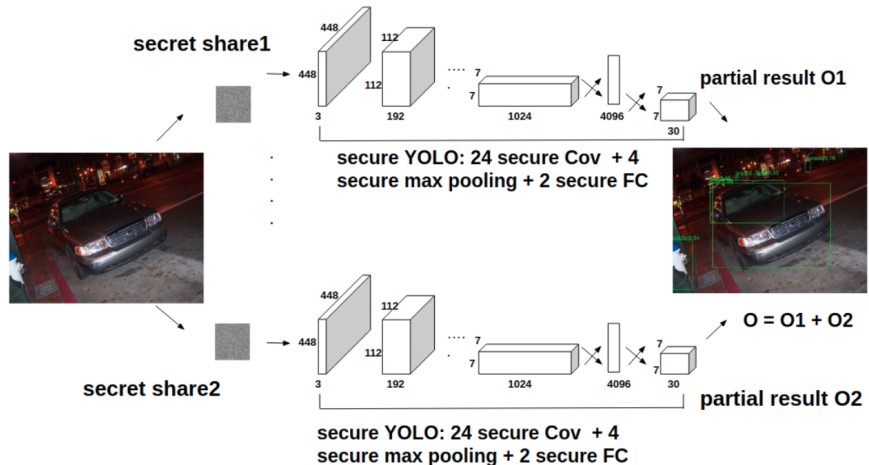

**Figure 3.** Structure and major components of vePOD YOLO. Image-level secret sharing is applied. Edge servers execute secure functions in vePOD YOLO to process secret shares.

A secret share $M_i$ received by an edge server goes through a number of secure convolutional layers, secure max-pooling layers, and secure fully-connected layers. Convolutional layers and fully-connected layers perform linear transformations that do not affect the additive relationship among secret shares. Thus, the secure convolutional layer and secure fully-connected layer are the same as the original convolutional and fully-connected layers, respectively. We focus on the design of the secure max-pooling layer.

Secure Function for Max-Pooling Layer

Max pooling keeps important features by selecting the largest, i.e., $max()$, in a stride. The $max()$ function performs a non-linear operation on features, which may affect the additive relationship among secret shares. To assure the max-pooling layer generates correct results on secret shares, we design a secure max-pooling function.

Algorithm 2 sketches the secure function for map pooling. In a vehicle–edge computing environment with two edge servers, secure map pooling works as follows.

An edge server $E_1$ processes a feature map $F_1 j$ in a channel $j$ and processes a region $R$ (e.g., $2 \times 2$) at a time. To find the maximum-valued element in the region $R$, it calculates the relative difference by subtracting the elements in $R$ from the two edge servers, exchanges

secret shares' relative difference, and adding them; i.e., if $I$ is smaller than zero, the value of the raw data element in the region $R$ with index $[\alpha, \beta]$ is smaller than the raw data element with the index $[w, h]$. The index of the greater element is recorded and this process is repeated until all the elements in $R$ are processed. Thus, the index of the maximum-valued element in $R$ can be determined. With this design, only the relative difference is exchanged between edge servers, which protects the feature maps and secret shares from being revealed to each other.

---

**Algorithm 2** Secure Max Pooling Function

---

1: Input: feature map $F_i$ from the previous convolution layer on edge server $E_i$
2: **for** each channel $j$ in feature map $F_i$ **do**
3:     $F_{ij}$ is the feature map $F_i$ in the $j_{th}$ channel
4:     $w, h$ are the max-pooling strides
5:     **for** each stride region $R$ in $F_{ij}$, $w$ in a range $[0, 1]$, $h$ in a range $[0, 1]$ **do**
6:         pooling index $\alpha = 0$, $\beta = 0$
7:         $I_i = R[\alpha][\beta] - R[w][h]$
8:         $E_i, E_q$ exchange $I_i, I_q$ and compute $I = I_i + I_q$
9:         **if** $I < 0$ **then**
10:             $\alpha = w, \beta = h$
11:         **end if**
12:         return $R[\alpha][\beta]$
13:     **end for**
14: **end for**

---

The secure max pooling in vePOD YOLO bears similarity to the secure ROI pooling in vePOD Faster R-CNN, but they are different. The former adopts region-based index tracking to find the most significant element, while the latter uses relative difference tables to select patch-based maximum-valued elements. Both methods exploit the additive secret sharing theory to enhance non-linear operations to handle secret shares.

## 3. Results

We implemented proof-of-concept vePOD CNNs, i.e., vePOD Faster R-CNN and vePOD YOLO, and evaluated their performance on a vehicular edge-computing testbed. Each edge server in the testbed is equipped with an AMD Ryzen 7 processor with 6 cores at 3.2 GHz and 16 GB DRAM, and runs Ubuntu Linux v20.04 and Python v3.8.

We conducted our experiments on the COCO dataset [25], which is a widely used object-detection dataset. Among the 80+ categories of objects in the COCO dataset, we are interested in the transportation-related objects, such as vehicles, traffic signs, and pedestrians. In our experiments, we build vePOD CNNs using pre-trained models, which are trained by using 5000 images from COCO with 135 epochs, 40 batches, and 0.01 learning rate. To evaluate the performance of vePOD networks, we selected 350 images from the COCO dataset that contain the 15 most frequent street view object classes in 10 batches for inference.

### 3.1. Accuracy of Object Detection

vePOD aims to enhance object-detection CNNs to process secret shares to protect vehicles' data privacy. The implementation of the secure functions should not affect the functional correctness of the object-detection network. In this set of experiments, we compared the detection accuracy of the vePOD CNNs with that of the original CNNs.

Figure 4 presents the detection results in single-object and multi-object scenarios. We find that the vePOD CNN detects all the objects in each image that the original CNN does, achieving the same accuracy. This implies that the structure and secure functions of vePOD do not alter the detection accuracy.

To quantitatively evaluate the detection accuracy, we calculated the error rate as the difference between the outputs from the vePOD CNN and those from the original CNN;

i.e., $|(d_{vePOD} - d_{CNN})/d_{CNN}|$, where $d_{CNN}$ and $d_{vePOD}$ are the object-detection output (includes classification score and bounding box's offset) from the CNN and vePOD CNN, respectively. In our experiments, the average error rate was within $e^{-7}$. That is, vePOD CNN has the same detection capability as the original CNN.

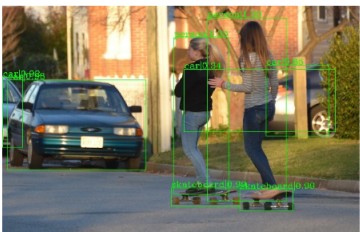
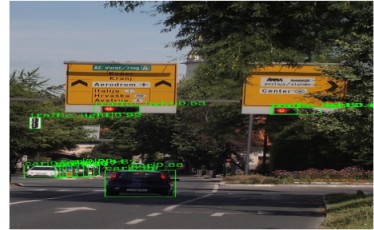
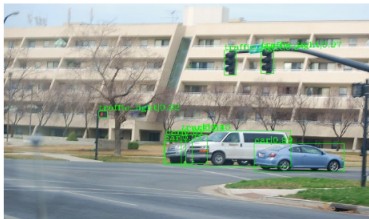
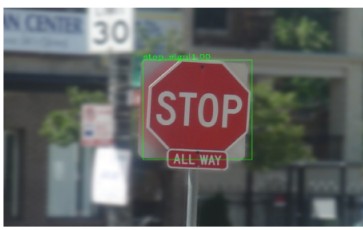

**Figure 4.** vePOD Faster R-CNN achieves the same detection accuracy as Faster R-CNN.

### 3.2. Security Analysis of vePOD

In vePOD, secret shares are sent to edge servers instead of raw images. These encrypted secret shares contain random data, which significantly enhances vePOD's resistance to cryptography attacks, such as eavesdropping attacks and chosen ciphertext attacks. Different ciphertext is generated even for the same plaintext each time. Furthermore, vePOD does not suffer from key management risks, since it divides and hides information in secret shares instead of through encryption.

In a vehicular edge-computing environment, edge servers may not be honest and they can collaborate to detect private information. To protect the privacy of a vehicle's sensor data, the vehicle can run one or more convolutional layers on a raw image locally to generate feature maps. Then, the secret shares of those feature maps are sent to edge servers where the remainder of the object-detection network will be executed. Even if edge servers collaborate with each other, they can only rebuild the feature maps. Thus, the original image data is protected.

### 3.3. Efficiency of Object Detection

#### 3.3.1. Performance of vePOD Faster R-CNN

We also evaluated the execution time of vePOD CNNs for detecting objects. Figure 5 shows the performance of vePOD Faster R-CNN and Faster R-CNN in processing $800 \times 800$ images.

For Faster R-CNN, the feature map generation using a backbone CNN takes 6.8 s, which accounts for 53.3% of the execution time. This is because the number of neural networks in the CNN backbone is more than five times the number in other parts of the detection network. VePOD Faster R-CNN takes a similar amount of time to generate feature maps and region proposals, as it uses the same CNN backbone and RPN network as Faster R-CNN.

Secret sharing uses storage and network bandwidth resources. We evaluated the transmission efficiency of vePOD. To transfer secret shares, vePOD Faster R-CNN takes 2.6 s under 300 Mbps bandwidth, which accounts for 5.9% of the overall execution time. In terms of the storage cost, we compare the size of secret shares with that of the original sensor data in Figure 10. Please see Section 3.3.4 for more details.

The major performance difference between vePOD Faster R-CNN and Faster R-CNN lies in the ROI pooling. Secure ROI pooling in vePOD Faster R-CNN takes 34 s, while Faster R-CNN uses 3.9 s.

This prolongation of the execution time is caused by the secure function, which calculates and merges a relative difference table for every region proposal specific patch.

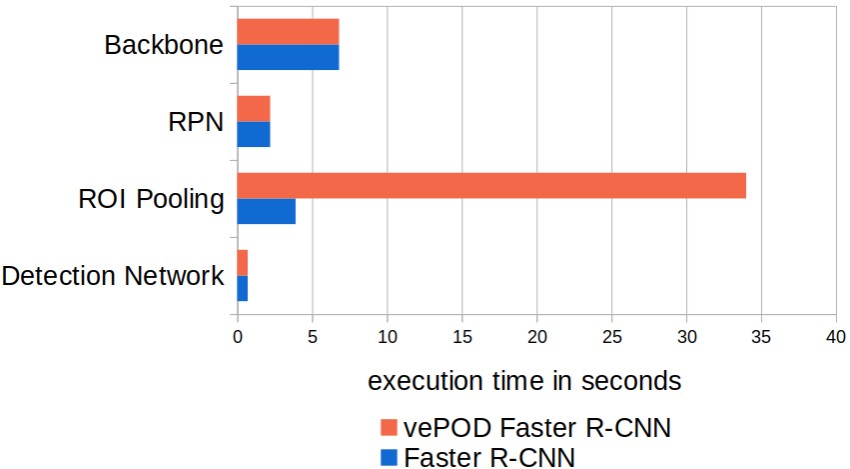

**Figure 5.** Comparison of the execution time of the key components in vePOD Faster R-CNN and Faster R-CNN.

To gain a deeper understanding of the preceding result, we study the impact of the number of region proposals on the performance of secure ROI pooling. Figure 6 presents the results. Faster R-CNN uses fixed-size input feature maps (50X50X512, i.e., width, height, and channel). The number of region proposals directly influences the execution time of the secure function. The time is comparable when there are 100 or 500 region proposals. Specifically, for 100 region proposals, the original ROI pooling takes 1.7 s and the secure ROI pooling uses 1.9 s. However, when there are over 1000 region proposals, the execution time differs more; i.e., the original ROI pooling takes 3.9 s to process 2000 region proposals. In contrast, the secure ROI pooling uses 34.0 s. In addition to the number of region proposals, the shape of the ROI affects the performance of the secure ROI pooling as well. More specifically, larger ROIs lead to an increased execution time caused by secure operations to construct the relative difference tables; for example, an ROI with size $12 \times 12$ (width, height) takes an additional 0.007 s computation time compared to an ROI with size $3 \times 3$.

Intuitively, when more secret shares are created from a vehicle's sensor data, vePOD Faster R-CNN achieves better privacy protection. However, this also increase the computation cost. Figure 6 plots the relationship between the execution time of the secure ROI pooling and the number of secret shares. Each secret share is processed by an edge node. The extra time caused by adding one more secret share is not significant, e.g., from 34.1 s for two secret shares to 37.1 s for three secret shares, an 8.1% increase. This is because most of the operations in the secure ROI pooling remain the same, except for adding an extra relative difference table from the additional secret share in every ROI patch.

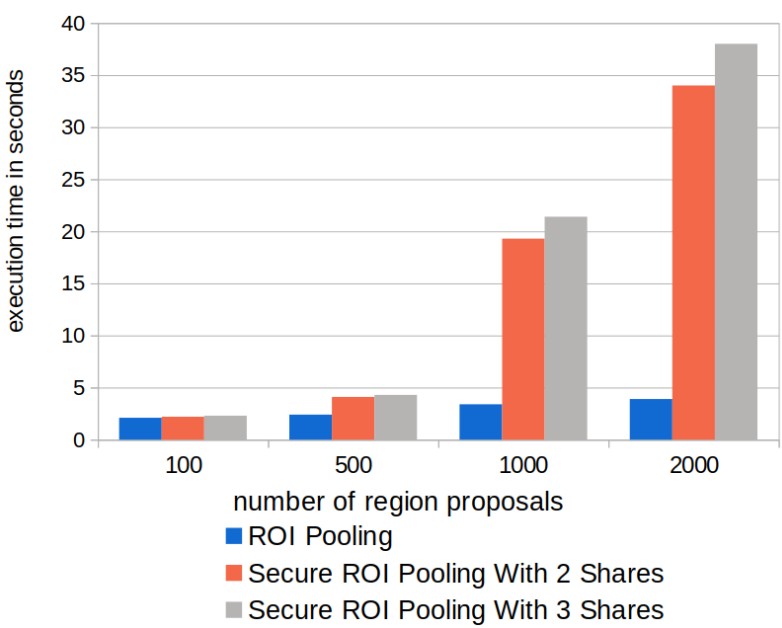

**Figure 6.** Execution time of the secure ROI pooling in vePOD Faster R-CNN for processing different numbers of region proposals.

### 3.3.2. Performance of Selective Secure ROI Pooling

The selective secure ROI pooling compares the indices of maximum-valued elements, which avoids calculating difference tables, to speed up the secure function of ROI pooling. Figure 7 presents the execution time of the selective secure ROI pooling. In the figure, we can see when there are two secret shares, the selective secure ROI pooling takes 29.7 s, achieving a speed-up of 14.6% over the secure ROI pooling. When the number of secret shares increases, the performance improvement becomes less significant. When the number of secret shares is three, the selective secure ROI pooling uses 40.3 s, that is, a performance degradation by 5.3% compared with the secure ROI pooling. The probability of successfully selecting the maximum-valued element at the same location among all the patches on different edge servers is $\frac{1}{(S_{ij})^k}$, where $S_{ij}$ is the patch size and $k$ is the number of secure shares. This probability drops when the number of edge servers increases.

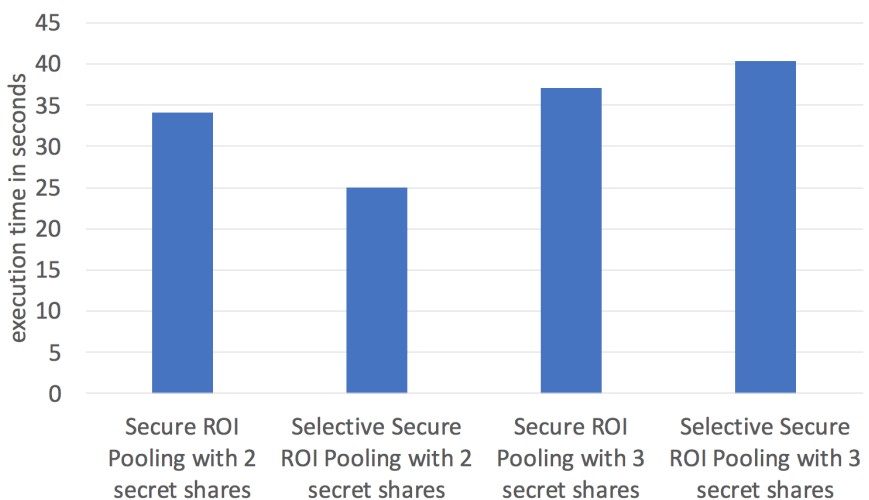

**Figure 7.** Performance comparison of secure ROI pooling and selective secure ROI pooling with two and three secure shares running vePOD Faster R-CNN.

### 3.3.3. Performance of vePOD YOLO

As a single-stage privacy-preserving object-detection network, vePOD YOLO processes secret shares through a series of secure convolutional, secure max pooling, and secure fully-connected layers without generating region proposals. We also evaluate our implementation of vePOD YOLO on the COCO dataset.

Figure 8 compares the performance of vePOD YOLO and the original YOLO using two and three edge servers. The dimension of input images to YOLO is the same as the dimension of image-level secret shares to vePOD YOLO. In the figure, we can see that the major performance difference between YOLO and vePOD YOLO is from the secure max pooling. Secure max pooling for two shares introduces 72.6 s computation latency and 4.1 s data-transmission latency, which takes 99.8% of total performance overhead.

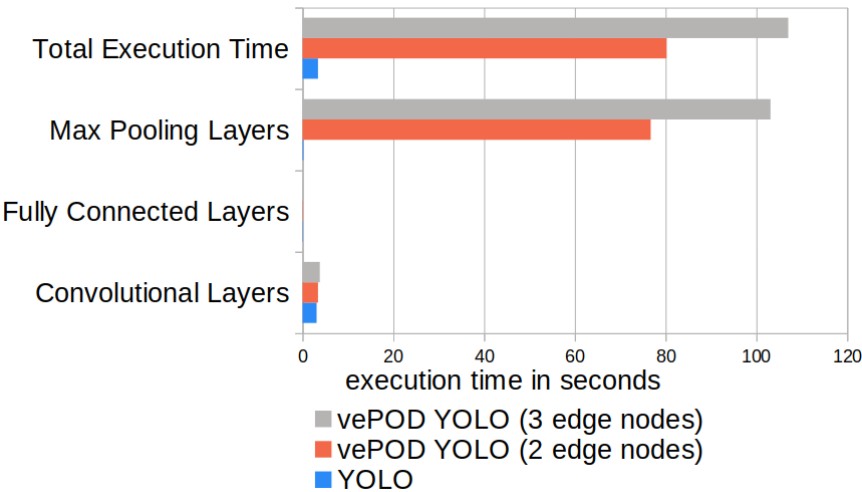

**Figure 8.** Performance comparison of YOLO and vePOD YOLO run on two and three edge servers.

The execution time of secure max pooling, as shown in Figure 9, increases linearly with the size of the input feature map. The size of a feature map equals the product of the width, height, and the number of channels of the feature map. For example, vePOD YOLO's execution time with two edge nodes is 15.17 s with input size $56 \times 56 \times 512$ (width, height, channel), and execution time for a feature map of input size $224 \times 224 \times 64$ is 32.4 s. The execution time ratio equals the ratio of the total number of features in the input feature map because the secure function is executed on every pooling region iteratively with identical time complexity. VePOD Darknet CNN consists of four secure max-pooling layers, and their execution time varies. The ratio of the execution time between the four layers matches the ratio of the corresponding size of their input feature maps. The ratio of the four layers' execution time and feature map size is 4:3:2:1. The execution time of vePOD YOLO with three nodes is longer than two nodes with the same input size. For example, with an input feature map size of $56 \times 56 \times 512$, the execution time for two and three nodes is 15.6 s and 19.2 s. Compared to two nodes, the secure function for three nodes needs to combine an extra piece of data to retrieve the difference of feature value in every location, which causes an additional delay in performance.

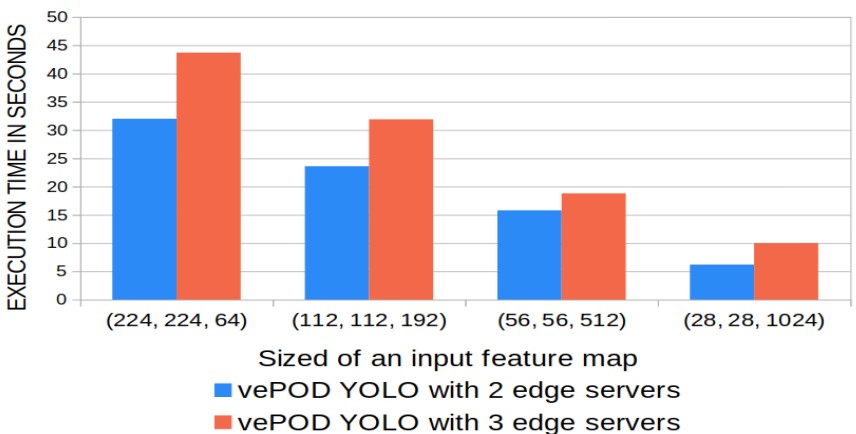

**Figure 9.** Execution time of the secure max pooling in vePOD YOLO with two and three edge servers.

### 3.3.4. Performance Comparison of vePOD Faster R-CNN and vePOD YOLO

Both vePOD Faster R-CNN and vePOD YOLO protect data privacy for vehicles and achieve the same detection accuracy as their original networks. Compared with vePOD YOLO, vePOD Faster R-CNN achieves a higher detection accuracy of around 8.4 percentage of MAP, especially for small clustered objects due to its two-stage detection approach.

Moreover, we find from the experiments that the execution time of vePOD YOLO increases more when $k$ becomes greater, e.g., from 2 to 3, compared with vePOD Faster R-CNN. This is because (1) in vePOD Faster R-CNN, when the number of secret shares $k$ increases, the secure ROI pooling becomes less complex compared with the secure max pooling in vePOD YOLO; and (2) the execution time increase of vePOD YOLO comes from the computation cost of the four secure max-pooling layers, while in vePOD Faster R-CNN, the execution time increase is from one secure ROI pooling layer.

Figure 10 shows that vePOD YOLO transfers less data between the vehicle and edge servers. This is because vePOD YOLO employs image-level secret sharing. In contrast, feature-level secret sharing needs to transfer both feature-map secret shares and region proposals from the vehicle to edge servers.

Secret shares need extra storage space. In Figure 10, we can see that vePOD Faster R-CNN on two edge servers for each vehicle's image needs 280 bytes of additional space to store secret shares compared to using the feature maps. VePOD YOLO under the same setting takes an extra 160 bytes compared to using the raw image.

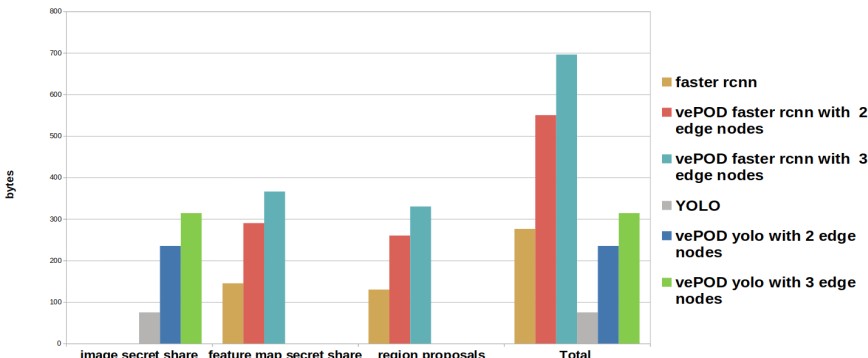

**Figure 10.** Comparison of storage and communication costs using vePOD Faster R-CNN and vePOD YOLO.

Both Figures 5 and 8 indicate that the secure functions prolong the execution of those layers containing nonlinear operations. To make vePOD more suitable for real-time object detection, there are several possible ways to improve its performance. For example, we can leverage process-level parallelism, e.g., using multi-threading, to speed up the

repeated/iterative invocations of a secure function; inside a secure function, we can identify and maximize data reuse to remove redundant computations and exploit instruction-level parallelism to shorten each invocation. Another main cause of the performance degradation is the use of Python code in the implementation. Currently, all the secure functions are written in Python. This eases the implementation, but at the cost of performance. We plan to rewrite the secure functions using lower-level programming languages, such as C or assembly, to speed up their execution, especially in nested loops.

## 4. Related Work

Privacy protection for deep learning is an important research topic and has attracted significant attention. A number of techniques have been proposed in the literature, such as homomorphic encryption and secure multi-party computation. In this section, we discuss the related research.

Leboe-McGowan et al. [6] proposed a heuristic privacy-preserving CNN for image classification. Instead of pursuing perfect ciphertext-based non-linear operations, they applied a rough approximation to evaluate the CNN's non-linear transformation layers. It achieved promising performance, since the complex ciphertext-based computation was avoided. However, the accuracy of object classification was compromised, with a 4% degradation of classification accuracy. Our vePOD targets the more complex object-detection tasks and it keeps the same detection accuracy as the original deep-learning networks.

Xie et al. [3] and Erkin et al. [4] applied homomorphic encryption to deep learning on network-connected servers. The former devised a secure CNN model to process encrypted data and generate results in cipher text that only the owner of the data can decrypt. The latter focused on secure image-classification CNNs using fully homomorphic encryption. These approaches aim to protect the input data and inference results. However, homomorphic encryption causes prohibitive computation overhead and drastic performance degradation. For example, an implementation of fully homomorphic encryption for deep learning suffered from a slowdown by four orders of magnitude [7]. The preceding works are for image classification. The performance degradation of homomorphic encryption for object detection is even worse, making it impractical for real-world deployment.

Secure machine learning [26] provides secure protocols for training ciphertext-based CNN models using linear regression and logistic regression. A privacy-preserving deep-learning framework [27] included secure protocols for connected servers to share model parameters. In the framework, no encryption or decryption was conducted on data, and there was no guarantee that adversaries could not use those crucial parameters to attack the system.

Object detection is vital for autonomous driving and is more complex than image classification. Privacy protection for object detection in a vehicle–edge environment has not been studied. In this paper, we tackle this new problem and present privacy-preserving object-detection networks that are run on edge servers to process secret shares from vehicles to achieve enhanced data privacy and uncompromised detection accuracy.

## 5. Conclusions

We present a privacy-preserving object-detection (vePOD) framework that can protect both a vehicle's sensor data and object-detection results from being exposed to and used by edge servers. We leverage the additive secret-sharing theory to develop secure functions in detection CNNs. The secure layers in a vePOD CNN process secret shares received from a vehicle and produce detections with the same functional correctness as the original network. We implement proof-of-concept networks for vePOD Faster R-CNN and vePOD YOLO. Experimental results on a public dataset show that vePOD CNNs guarantee the same detection accuracy as the original CNNs. Most importantly, vePOD CNNs protect data privacy for a vehicle by assuring that each edge server only accesses an encrypted secret share and produces partial detection, which has no semantic importance for the edge server.

In our experiments, we note the performance degradation in running vePOD CNNs, especially from the secure ROI pooling. The main goal of this paper is to develop proof-of-concept networks for privacy-preserving object detection in a vehicle–edge collaborative environment. We did not conduct any optimization on it. In our future research, we will speed up the execution of vePOD CNNs by exploring acceleration and parallel-processing techniques. Furthermore, we plan to develop novel approaches to enable the execution of the entire object-detection workflow on edge servers for all types of CNNs. This will free the limited resources on a vehicle to perform more mission-critical tasks.

**Author Contributions:** Supervision, S.F. and Q.Y.; Writing—original draft, T.B.; Writing—review & editing, S.F. and Q.Y. All authors have read and agreed to the published version of the manuscript.

**Funding:** This work has been supported in part by the U.S. National Science Foundation grants CNS-2037982, ECCS-2010332, OAC-2017564, CNS-2113805, CNS-1852134, DUE-2225229, and CNS-1828105. We thank the reviewers for their constructive comments and suggestions, which helped us improve this paper.

**Institutional Review Board Statement:** Not applicable.

**Informed Consent Statement:** Not applicable.

**Data Availability Statement:** Not applicable.

**Conflicts of Interest:** The authors declare no conflict of interest.

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
