# Peer review of "Privacy-Preserving Object Detection with Secure Convolutional Neural Networks for Vehicular Edge Computing"

_futureinternet, doi:10.3390/fi14110316_

Round 1

Reviewer 1 Report

This paper presents a privacy-preserving object detection (vePOD) framework that can protect both vehicle’s sensor data and object detection results from being exposed to and used by edge servers. The paper is well organized and easy to follow. The authors clearly introduced the proposed approach along with associated technical content. The breakdown of structure and components of the VePOD framework. The experimental results showed the accuracy and efficiency of  Overall, the paper is in good form. Some minor interrogative concerns:

1. The core idea for security is the additive secret sharing. The disadvantage  of secret sharing is the required storage and bandwidth resources. The authors showed the testing result in processing time. The storage and transmission efficiency may need some discussions as well. 

2. The paper is to design a more secure and privacy-preserved approach.  The Result section mainly showed the accuracy and efficiency. Although the proposed method enabled the security, an analysis of the security of the approach should be provided. 

Author Response

Summary of Revision

We would like to thank all reviewers for their constructive comments and suggestions. We have revised our manuscript and addressed all the comments. We believe the revised version is much stronger. The following changes were made in the revision.

Reviewer #1:

Review comment 1: The core idea for security is the additive secret sharing. The disadvantage of secret sharing is the required storage and bandwidth resources. The authors showed the testing result in processing time. The storage and transmission efficiency may need some discussions as well. 

Response: We added the results of network transmission time and bandwidth to Section 3.3.1. 
With regard to the storage requirement, in Figure 10, we added the data sizes from the original Faster R-CNN and YOLO without secure functions to show the additional storage space needed for secret sharing.

Review comment 2: The paper is to design a more secure and privacy-preserved approach.  The Result section mainly showed the accuracy and efficiency. Although the proposed method enabled the security, an analysis of the security of the approach should be provided. 

Response: We added a new section, Section 3.2, to analyze the security enhancement by using vePOD. We also discuss the case where edge servers collaborate to discover vehicles’ data and present a possible solution.  

Reviewer #2:

Review comment 1: It is better to describe a specific application scenario where and how the proposed technology can be applied. 

Response: We added an application scenario in Section 2.3. In the scenario, two or more vehicles exchange sensor data for cooperative perception which helps improve the perception range and accuracy. To prevent a vehicle from being overloaded, object detection on images from other vehicles need to be performed on edge servers. The vePOD can be applied to protect the data privacy for vehicles. We described how to apply vePOD in Section 2.3. 

Review comment 2: The paper assumes edge nodes are honest and they will not collaborate to detect private information. Authors can add some discussion on the scenario where these edge nodes cannot be fully trusted.

Response: We added a new section, Section 3.2, to analyze the security of vePOD. Specifically, we discuss the scenario where edge servers may collaborate. To protect data privacy, a vehicle can run convolutional layers on the image data to generate feature maps. Then the secret shares of those feature maps are sent to edge servers where the reminder of an object detection network is executed. Even if edge servers collaborate with each other, they can only rebuild the feature maps. Thus, the original image data is protected.  

Review comment 3: It seems the proposed algorithm introduces heavy overheads. Can the authors add some discussion about how to reduce the overheads?

Response: We added discussion about possible ways to improve the performance in Section 3.3.4. These include exploiting process-level parallelism (e.g., multi-threading) between calls to the secure functions, instruction-level parallelism and data reuse within a secure function, and lower-level programming languages (such as C or assembly) to speed up vePOD networks.      

Review comment 4: More explanations are expected on how to define and detect ROIs. 

Response: We added the definition and detection of ROIs in Section 2.5. 

Review comment 5: More information about the dataset can be added, especially how the dataset is related to and is used in the experiment.

Response: We used the COCO dataset in our experiments. The dataset contains images with objects of cars, traffic signs, and pedestrians. We used those images in training and inference. We added the detail of the dataset and how we used it in Section 3. 

Reviewer 2 Report

This manuscript proposes a privacy-preserving object detection framework that leverages faster R-CNN, YOLO, and edge computing to preserve privacy while sharing data secretly among multiple autonomous vehicles.  

The paper is well-written and well-organized. The ideas presented in the paper have novelty and are interesting. However, the paper may be improved as follows. 

1) It is better to describe a specific application scenario where and how the proposed technology can be applied. 

2) The paper assumes edge nodes are honest and they will not collaborate to detect private information. Authors can add some discussion on the scenario where these edge nodes cannot be fully trusted.

3) It seems the proposed algorithm introduces heavy overheads. Can the authors add some discussion about how to reduce the overheads?

4) More explanations are expected on how to define and detect ROIs. 

5) More information about the dataset can be added, especially how the dataset is related to and is used in the experiment.

Author Response

(The authors gave the same response as above.)
